# Fires in Waste Treatment Facilities: Challenges and Solutions from a Fire Investigation Perspective

**Wen-Yen Juan [1], Chia-Lung Wu [2],\*, Fan-Wei Liu [3] and Wei-Sheng Chen [1],\***

1   Department of Resources Engineering, National Cheng Kung University, Tainan 701401, Taiwan; n48111075@gs.ncku.edu.tw
2   School of Safety and Health Sciences, Chang Jung Christian University, Tainan 71101, Taiwan
3   Academy of Circular Economy, National Chung Hsing University, Nantou 540216, Taiwan; fwliu@dragon.nchu.edu.tw
\*   Correspondence: farianwu@mail.cjcu.edu.tw (C.-L.W.); kenchen@mail.ncku.edu.tw (W.-S.C.)

**Abstract:** Fires in waste treatment facilities have significant social, economic, and environmental implications. Factors such as self-heating of lithium-ion batteries, thermal runaway, friction, human activities, technical errors, and unfavorable storage conditions contribute to these fires. High-risk categories include illegal dumping sites, recycle collection stations, and wood-related facilities. The frequency of fires in waste treatment facilities and the emergence of new waste types have led to public discontent. Regulatory challenges and oversight difficulties pose further obstacles. This study analyzes fire incidents in Kaohsiung City's waste treatment facilities over the past five years, exploring their causes, regulatory frameworks, and practical challenges. Valuable insights and recommendations are provided to enhance fire safety and risk assessment. These measures aim to mitigate fires' environmental and facility consequences and their impact. Prioritizing fire prevention and reducing potential economic, social, and environmental consequences are crucial for improving fire safety in waste treatment facilities. Addressing these challenges and prioritizing the safety and sustainability of the waste treatment industry is imperative.

**Keywords:** waste treatment and management; waste fire; fire safety; cause of fire; environmental impact

## 1. Introduction

As the population grows and science and technology advance, the volume of waste produced continues to escalate. According to statistics, the Taiwan region generated more than 10 million tons of general waste in 2021 [1]. This surge in general waste has necessitated an increase in waste treatment facilities such as incinerators, landfills, and temporary storage sites. On the other hand, unlike general waste, industrial waste has also risen due to economic growth, demanding more careful handling. Consequently, the treatment cost for industrial waste is higher than for general waste, prompting some unscrupulous businessmen to dump industrial waste haphazardly. This contributes to environmental pollution and poses a risk of arbitrary or spontaneous ignition if the industrial waste is flammable. Consequently, fires at these facilities have become a pressing issue. While discussions on the environmental impact of fires traditionally focused on the emissions that they release into the air, water, and soil, it is crucial to recognize that fires in waste have ecological, economic, and social repercussions [2]. These consequences can have severe implications for society as a whole.

From the point of view of firefighters, fighting waste treatment facility fires is a daunting challenge. Workers typically compress waste into piles to save limited space in most waste disposal processes. However, in the event of a fire, it tends to develop into a deep-seated fire, which, combined with the high fire load of the piles of waste, will require extensive watering over a lengthy period to put it out. However, there has been a notable increase in significant fires at these facilities in recent years. For instance,

in 2018, a wastepaper recycling plant in Daliao District, Kaohsiung City, in the Taiwan region experienced a fire that took firefighters an entire week to extinguish completely. The incident resulted in significant damage and losses, necessitating the deployment of over 300 firefighters and utilizing approximately 20,000 tons of water [3]. The affected facility suffered a loss exceeding NTD 50 million and was fined NTD 5 million for the air pollution. These numbers far exceed the average resources used for general building fires (typically about 20–30 firefighters and 90–120 tons of water) [4], indicating the severity and scale of the incident. Furthermore, the same site witnessed two to three consecutive fires in the subsequent years, indicating the urgent need to prioritize fire safety measures and highlighting a long-standing disregard and neglect for fire safety and risk management in these facilities.

Furthermore, the air monitoring report revealed alarming dioxin emission levels during the waste treatment facility fire, with the emission coefficient ranging between 4.4 and 5000 ng-iTEQ/kg. These values far exceed the current regulatory limit of 0.5 ng-iTEQ/kg, significantly harming the surrounding environment and its inhabitants [5]; the implications for health and ecology are substantial. Consequently, prioritizing fire prevention and implementing robust risk management measures in waste treatment facilities are crucial considerations.

Numerous scholars have initiated discussions and conducted studies on fires in waste facilities in recent years. Ibrahim et al. [6] explored the possible social, health, and environmental consequences of fires at temporary storage sites for municipal solid waste in Sweden. The study considered factors such as population density and the location of the storage sites. The researchers concluded that the frequency of fires and the risk associated with privately managed storage sites were significantly higher than government-mandated storage sites. Additionally, studies found that the average population residing within a radius of 1–3 km from these storage sites also faced increased risks. These findings shed light on the inappropriate siting of temporary municipal solid waste storage sites. Nigl et al. [7] conducted a comprehensive investigation and analysis of fire incidents in waste facilities in Austria spanning the last decade. Their study delved into specific patterns of waste fires, including cases of spontaneous ignition and the increasing array of potential ignition sources. The study identified statistical correlations between the probability of fire events and seasonal or climatic factors. Furthermore, the researchers compared the occurrence and trends of waste facility fires with those in neighboring countries. This study provided valuable insights into the characteristics and dynamics of waste facility fires, offering a broader perspective on the subject. Mikalsen et al. [2] studied fires at waste treatment facilities in Norway and Sweden, conducted a comprehensive fire safety assessment, and drew conclusions on improving fire safety. Their study also recommends the design, operation, waste handling, and storage of fire safety instruments, and efforts to limit the consequences on the environment and facilities during and after a fire.

Simultaneously, numerous scholars have conducted research on specific waste treatment facility fires. For example, Sotto et al. [8] and Mocellin et al. [9] conducted numerical simulations and analyzed the development of fires in waste treatment plants, providing insights into fire safety design standards. Caetano et al. [10], Diaz et al. [11], and Li et al. [12] discussed the hazards in the recycling process of electronic waste, particularly in the recycling of lithium batteries. Gallo et al. [13] addressed site selection and safety issues in landfills. Ibrahim et al. [14] and Gogola et al. [15] focused on the spontaneous combustion of organic waste and storage sites for extractive waste. These studies contribute to a deeper understanding of fire risks and provide valuable insights for fire prevention and safety in specific waste treatment facility contexts.

Many scholars have also conducted research on specific ignition sources and mechanisms. For example, Browne et al. [16], Gray et al. [17], Ibrahim et al. [18], Li et al. [19], Blijderveen et al. [20], and Wang et al. [21] have studied the spontaneous combustion, ignition mechanisms, and fire risks of wood, hay, and fiber waste. Lisbona et al. [22], Nigl et al. [23,24], and Velázquez-Martínez et al. [25] have discussed the ignition mecha-

nisms of lithium batteries, such as thermal runaway or internal short circuits, aiming to identify ways to prevent disasters during the recycling, storage, and handling processes. Martin et al. [26] have explored the hazards associated with the disposal of aluminum production waste in landfills and proposed monitoring methods for early fire detection, facilitating effective response measures.

Despite the increasing attention from scholars worldwide towards fires in waste treatment facilities, there is a noticeable lack of systematic research in the Taiwan region regarding the causes, prevention, risk assessment, classification, and necessary safety measures associated with these incidents, and the challenges are further compounded by difficulties in accessing relevant databases and issues with statistical classification. Analyzing past fire cases becomes imperative to gain a comprehensive understanding of fires in waste treatment facilities and effectively assess the associated risks. The following aspects can be explored by studying these cases and capitalizing on the opportunity to enhance facility risk management in this field.

1.  Most common causes of fires: Investigating the primary causes of fires in waste treatment facilities, such as the ignition sources, human activities, equipment malfunctions, or environmental factors that contribute to fire incidents.
2.  Timing and locations of fire incidents: Analyzing when and where the fires at these waste treatment facilities typically occur. This includes identifying patterns in terms of seasons, weather conditions, and specific facility characteristics that may increase the likelihood of fire incidents.
3.  Factors affecting fire occurrence: Understanding the factors that influence the occurrence of fires in waste treatment facilities. This can involve studying the impact of weather conditions and human behaviors on fire risks and the behavior of waste materials.
4.  Safety assessment of waste treatment facilities: Conducting a comprehensive safety assessment of waste treatment facilities, including evaluating the design, operation, waste handling practices, fire prevention measures, emergency response plans, and overall fire safety infrastructure in place.
5.  Safety advice measures for related facilities: Providing practical recommendations and measures to enhance the fire safety of waste treatment facilities. This may involve implementing proper waste management protocols, training staff on fire prevention and response, ensuring the presence of adequate fire suppression systems, and conducting regular inspections and maintenance of equipment and facilities.

## 2. Methodology

For this study, three approaches were taken. First, a review was conducted on the preliminary rescue reports of waste facility fires among all fires in Kaohsiung City between 2017 and 2021. This allowed for examining the specific details and circumstances surrounding these incidents. Second, on-site interviews were conducted with firefighters with firsthand experience in waste fires. These interviews provided valuable insights into the challenges faced, strategies employed, and lessons learned during firefighting operations at waste facilities. Last, a cross-comparison was drawn between the rescue reports and relevant investigation reports or records. This approach enabled a comprehensive analysis of the data, facilitating the identification of patterns, correlations, and potential areas for improvement in waste facility fire response and prevention. By employing these three approaches, the study aimed to gain a comprehensive understanding of waste facility fires in Kaohsiung City, contributing to developing effective strategies and measures for fire prevention, response, and mitigation in such facilities.

In the first approach, a selection was made from a total of 21,183 fire incidents that occurred in Kaohsiung City between 2017 and 2021. The specific locations of interest for this study were as follows: (1) Incinerators; (2) Recycle collection stations; (3) Temporary waste storage sites; (4) Illegal dumping sites; (5) Wood-related facilities; (6) Landfills; (7) Industrial waste treatment plants; (8) Recycle treatment plants; and (9) Transport vehicles. However,

some facilities are complex objects of two or more types and are classified according to the main operating object.

From these selected incidents, the following information was collected: (1) date and time of the fire; (2) location of the fire; (3) affected area; (4) the operation status of the facility; and (5) rescue duration. Historical meteorological data were also gathered to obtain the weather conditions and temperature during each fire incident. This comprehensive dataset allows for a detailed analysis of the relationship between waste facility fires and various factors such as location, weather, and operational conditions.

In the second approach, after screening fire cases related to waste treatment facilities, the open-ended question interview was conducted with 45 firefighters from 20 fire stations who were directly involved in the rescue operations of selected fire accidents (with some firefighters having experience in more than two of the above-mentioned fire accidents). The purpose of the interviews was to gather detailed information and insights that were not recorded in the rescue reports to gain a better understanding of the specific details of each fire incident. During the interviews, firefighters were asked about various aspects related to the waste fires, including (1) how the waste was stacked or stored; (2) the presence of different types of waste materials; (3) the situation under which the fire was burning; (4) the impact on individuals or people in the vicinity; (5) any available photos or videos taken during the fire incident; and (6) the deficiencies of on-site fire management and suggestions. These interviews yielded valuable first-hand information and detail, supplementing information that some rescue reports lacked, thus allowing for a more comprehensive analysis of each waste fire case. This approach also helped to ensure that only relevant and accurate fire incidents were included in the study, thereby ruling out any potential mistakenly selected cases.

The research in this study defined its system boundaries as follows:

- Spatial boundary: Kaohsiung City.
- Temporal boundary: Five years, from January 2017 to December 2021.
- Data availability: Rescue reports (approach 1), interview data (approach 2), and investigative reports and records.

For each fire case included in the study, the following data and parameters were collected:

- Date of the fire incident (additional derived variables: weekday, month, year, and season).
- Administrative region and approximate address (with privacy protection measures).
- Type of facility where the fire occurred.
- Type of waste involved in the fire.
- Cause of the fire.

## 3. Results and Discussion

After examining the 21,183 fire incidents that occurred in Kaohsiung City between 2017 and 2021 and excluding cases such as farmers burning leaves and hay, as well as fire cases with unclear information, a total of 299 waste-related fires were identified during this period.

### 3.1. Geographical Distribution Pattern

The geographical distribution of the 299 fire incidents in Kaohsiung City indicates that most waste-related fires occurred in areas with a medium population density; see Figure 1. This suggests that waste treatment facilities are predominantly located in medium population density areas surrounding densely populated regions. Two main reasons contribute to this pattern:

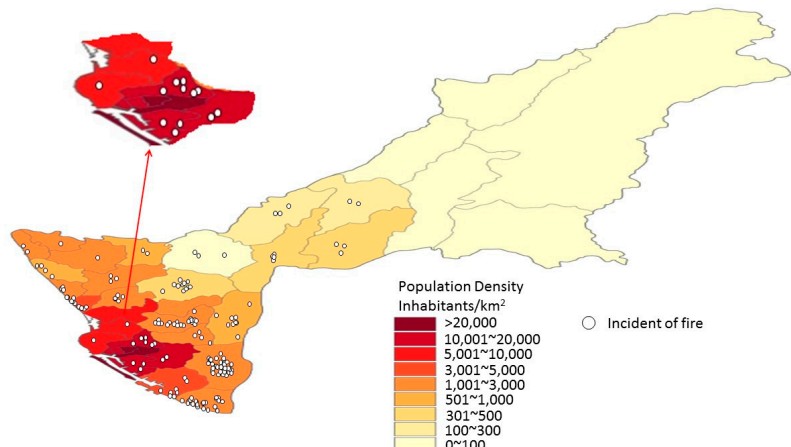

**Figure 1.** Geographical distribution of population density and waste facility fires in Kaohsiung City (2017~2021).

Not in My Backyard (NIMBY) Syndrome: Waste treatment facilities are often associated with unpleasant odors and mess, leading to a general reluctance of people to have such facilities near their residences. This phenomenon is known as the NIMBY syndrome. As a result, waste treatment facilities are typically situated away from densely populated areas.

Optimization of Waste Transportation Cost: Higher population density areas tend to generate more waste. Placing waste treatment facilities too far away from areas with high population density would increase transportation costs. Thus, waste treatment facilities are strategically located in industrial areas close to urban or residential regions, or even abandoned farmland, to optimize efficiency and reduce transportation expenses.

Consequently, it can be observed that areas with a relatively high density of fire incidents are mostly found in industrial areas. In sparsely populated regions, waste fires are often linked to illegal dumping sites, as identified through site photos and investigation reports. Understanding the geographical distribution of waste-related fires provides valuable insights into the location of waste treatment facilities, the impact of the NIMBY syndrome, and the importance of waste transportation optimization. These findings contribute to developing fire prevention and management strategies in waste facilities.

### 3.2. Temporal Distribution Pattern

Figure 2 shows the arrangement of 299 waste facility fires by month, along with a comparison with the monthly average rainfall and temperature. Based on the analysis of waste-related fires in Kaohsiung City and the comparison with average monthly precipitation over the past five years, several findings can be highlighted:

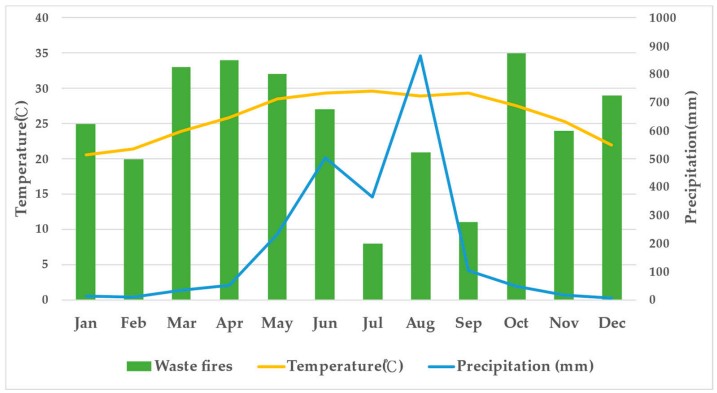

**Figure 2.** Monthly Fire Incident Distribution and relative comparison with precipitation and temperature.

Impact of Temperature: While temperature still influences the occurrence of waste-related fires to some extent, the analysis suggests that the effect of temperature is relatively small compared to the influence of precipitation. Rainfall is considered to be more important in determining the likelihood of fires at waste facilities. This is because precipitation helps to reduce the dryness and flammability of waste materials, thus contributing to controlling fire events.

Impact of Season and Precipitation: The influence of seasons on waste-related fires in Kaohsiung City can be observed in Figure 2. It shows that there are higher occurrences of fires related to waste between March and May and between October and December compared with other seasons. This pattern is due not only to the drier weather with less precipitation on average during this period but also to the agricultural and traditional religious practices that prevailed during these times. From March to May, farmers engage in spring ploughing activities. Before ploughing, it is common for farmers to burn weeds as a method of land preparation. This practice serves the dual purpose of reducing the amount of garbage and using the ashes as fertilizer. This period coincides with traditional ancestor worship practices where gold paper is burned as offerings. These burnings take place in proximity to agricultural areas. As mentioned above in the discussion on geographical distribution, some abandoned farmlands are used as locations for illegal waste dumping. If weeds or gold paper are burned nearby, sparks carried by the wind can potentially ignite nearby waste piles due to the dryness caused by a lack of rainfall. The occurrence of waste-related fires between October and December can also be attributed to farmers' agricultural practices. After the autumn harvest, farmers often burn straw and branches as a means of disposal. This agricultural burning activity, combined with the presence of waste materials, can create conditions that increase the risk of fires during this period.

These findings underscore the significance of considering weather conditions, particularly precipitation, when examining waste-related fires' occurrence and temporal distribution. However, it is important to note that other factors, such as human-induced ignition sources, may also play a role in fire incidents and must be considered for a comprehensive understanding.

### 3.3. Fire Investigation

The data from the 299 waste-related fires were analyzed to provide a breakdown based on the type of facility from where the fire originated and the cause of the fire. See Table 1 and Figure 3.

**Table 1.** The distribution of cause of fires vs. types of waste disposal facility.

| | Open Flame | Mechanical Energy | Electrical Factor | Welding Spark | Spontaneous Fire | Cigarette | Lithium Battery | Unknown | Re-Ignition |
|---|---|---|---|---|---|---|---|---|---|
| **Incinerators** | 0 | 1 | 0 | 1 | 0 | 0 | 2 | 0 | 0 |
| **Recycle collection station** | 18 | 4 | 5 | 7 | 0 | 18 | 6 | 18 | 0 |
| **Temporary waste storage site** | 0 | 0 | 0 | 1 | 0 | 4 | 1 | 1 | 0 |
| **Illegal dumping site** | 10 | 0 | 0 | 1 | 7 | 26 | 0 | 29 | 9 |
| **Wood-related** | 3 | 3 | 1 | 0 | 20 | 23 | 0 | 26 | 7 |
| **Landfill** | 0 | 0 | 1 | 0 | 2 | 0 | 0 | 0 | 0 |
| **Industrial waste treatment plant** | 1 | 4 | 0 | 1 | 3 | 1 | 0 | 1 | 0 |
| **Recycle treatment plant** | 3 | 5 | 0 | 10 | 5 | 2 | 2 | 2 | 0 |
| **Transport vehicle** | 0 | 0 | 0 | 0 | 0 | 2 | 1 | 1 | 0 |

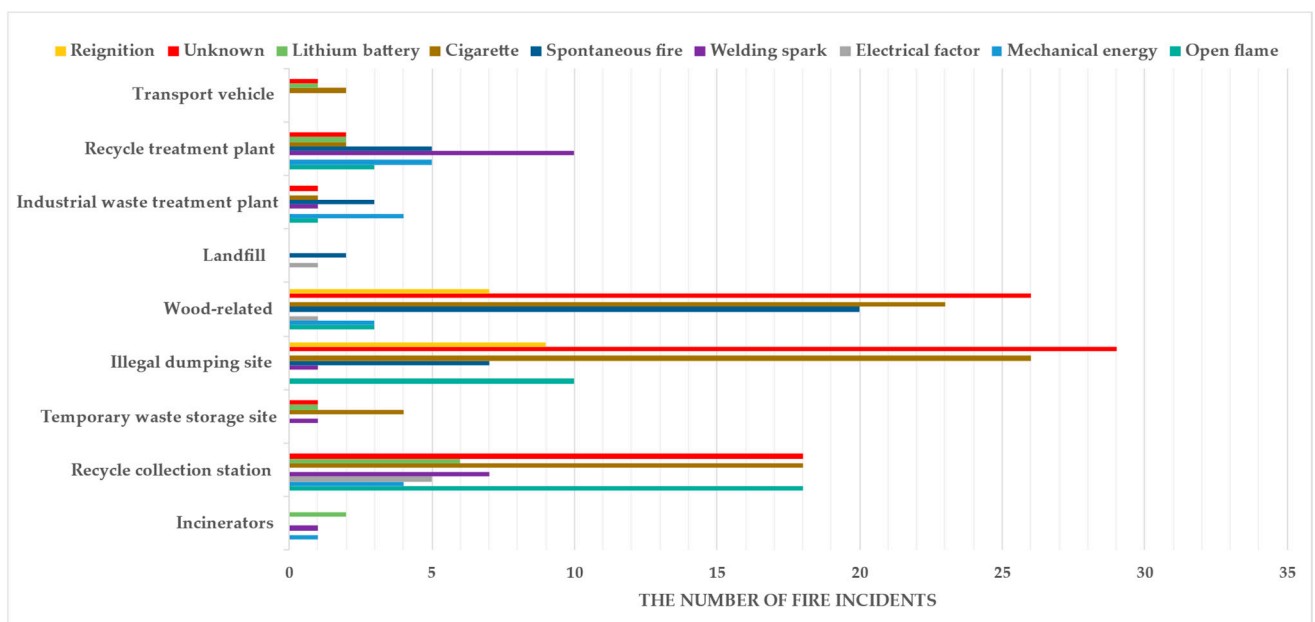

**Figure 3.** The distribution of cause of fires for different types of facility.

### 3.3.1. Discussion in Facilities

Based on the data provided in Table 1, it is evident that waste treatment facilities categorized as 'wood-related', including those involved in waste wood shredding and wood chip stockpiling, the 'illegal dumping sites' and 'recycle collection sites' were more susceptible to fire incidents compared to other types of waste treatment facilities. Consequently, these specific categories of waste treatment facilities have a higher probability of experiencing fires. Furthermore, a larger proportion of these fires are attributed to causes such as cigarettes, open flames, and unknown factors. Additionally, 'wood-related' facilities and illegal dumping sites also exhibit a higher occurrence of spontaneous and re-ignition fires. The cause of fire in these facilities, except for the cases classified as "unknown", which indicates insufficient evidence to determine the exact cause, can be analyzed from various perspectives.

Factors such as human behaviors and negligence, including improper disposal of cigarette butts or the presence of open flames where they should not be allowed, may contribute to fires caused by cigarettes and open flames. Enhancing regulations, implementing appropriate signage, and enforcing safety measures are crucial to prevent smoking-related fires and fires caused by open flames in these facilities. The classification of fires as "unknown" emphasizes the importance of thorough investigation and documentation to enhance understanding and identify potential causes. It also underscores the need to improve fire detection and monitoring systems in these facilities to aid in accurately determining the causes of fires.

The higher proportion of spontaneous and re-ignition fires in 'wood-related' facilities and illegal dumping sites suggests specific factors contributing to these incidents. In 'wood-related' facilities, factors such as heat buildup, improper storage, or handling of wood waste, and the presence of flammable materials can lead to spontaneous combustion. Similarly, in illegal dumping sites, the accumulation of combustible waste and improper disposal practices can result in fires that may reignite due to smoldering materials or the rekindling of flames. Preventive measures such as proper storage practices and regular monitoring are essential to mitigate these risks.

Analyzing the causes of fires in 'wood-related' facilities, 'illegal dumping sites', and 'recycle collection sites' from different perspectives provides a comprehensive understanding of the contributing factors. This knowledge enables the development of targeted

strategies to address these specific challenges and improve fire prevention measures in these vulnerable waste treatment facilities.

3.3.2. Discussion on Cause of Fires

Based on the data presented in Figure 4, it is notable that the most frequent cause of fire incidents is classified as "unknown", accounting for a quarter of the cases. This designation indicates that investigators were unable to gather conclusive evidence at the fire scene to support any specific hypothesis during the investigation. Such instances often arise when there is excessive waste accumulation on the site, making it challenging to clean up and find evidence. Additionally, the lack of on-site monitoring or security systems, absence of witnesses to the initial fire situation, or the presence of illegal dumping sites with few people can contribute to this classification.

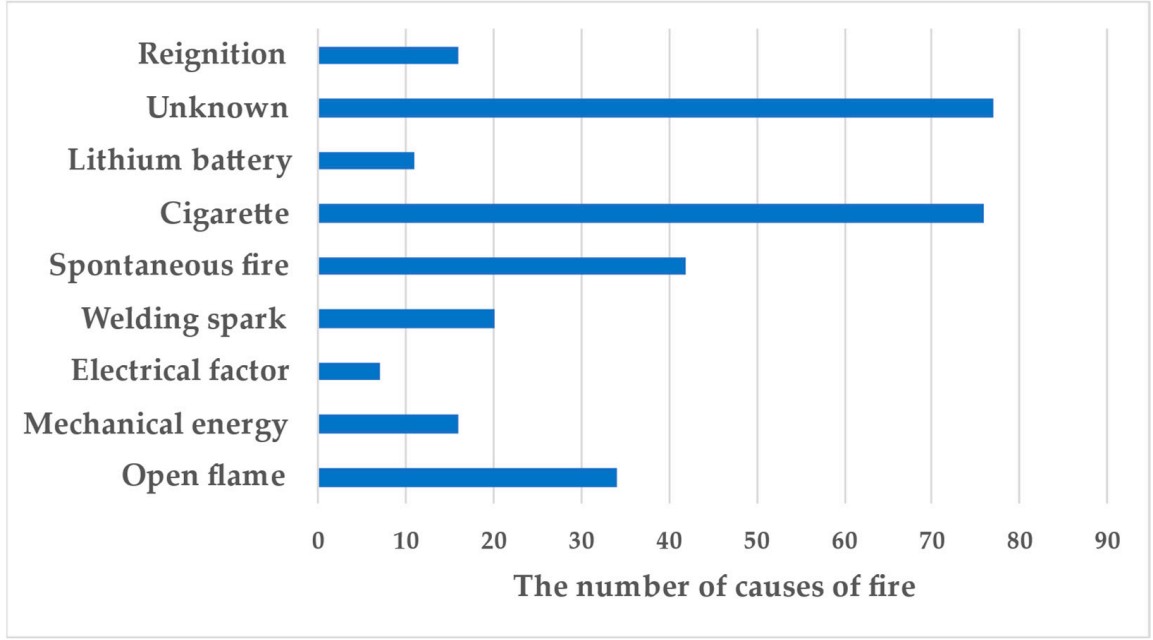

**Figure 4.** The statistics of fire causes.

The second most prevalent cause of fire incidents, also accounting for a quarter of the cases, is attributed to cigarettes. Fire incidents caused by discarded cigarette butts predominantly occur in "recycle collection stations", "illegal dumping sites", and "wood-related areas". The cause is that these facilities can be mistaken for mere "garbage dumps", leading to frequent instances of passersby indiscriminately disposing of cigarette butts, thereby increasing the likelihood of fires in these areas. Another situation arises from poor management, where smoking is not strictly prohibited or designated smoking areas are not established within the facility. As a result, employees smoking during work can result in cigarette butts or ash falling onto combustible waste, consequently causing fires.

Among all the fire causes, except for "unknown" and "spontaneous combustion", which are relatively related to natural and storage environmental conditions, other fire causes are mostly related to human factors, such as "cigarettes", "open flame", and "welding spark", etc. Therefore, it is important to note that human factors play a crucial role in these fire causes as well. For example, the improper handling of open flames or failure to adhere to safety protocols during welding operations can lead to fire incidents. This highlights the significance of proper training, adherence to safety guidelines, and responsible behavior in preventing fires associated with open flames and welding sparks. Overall, while natural environmental conditions can contribute to fire incidents in some cases, many of the other fire causes, such as cigarettes, open flames, and welding sparks, are closely tied to human actions and behaviors. Therefore, promoting awareness, implementing appropriate safety

measures, and fostering responsible practices are essential in minimizing the risk of fires in waste treatment facilities.

### 3.4. Fire Rescue

#### 3.4.1. Rescue Duration

Figure 5 provides statistics on the average time taken by firefighters to combat various types of waste treatment facility fires relating to the types of waste treatment facility fires whose rescue duration exceeds the total average of 310 min are marked in orange. The data reveal that, among the types of waste treatment facilities that experienced fires, the ones that required an average rescue time of over 8 h were "wood-related", "landfills", and "recycle treatment plants". Additionally, it is worth noting that incinerator fires had an average rescue time of nearly 6 h. This can be attributed to the location of the fire within the waste collection area of the incinerator, where a significant amount of municipal solid waste accumulates, making fire extinguishment more challenging.

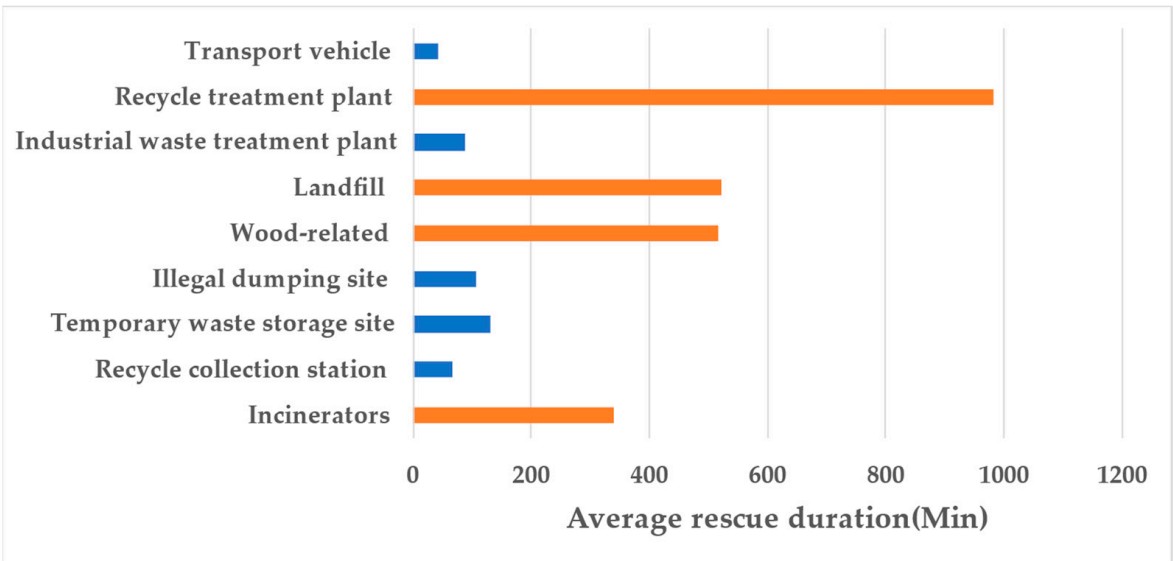

**Figure 5.** The average rescue duration of each type of facility.

The prolonged duration of firefighting operations contributes implications. It can significantly impact the damage sustained by the facility's equipment, as well as its effects on the environment and society. Therefore, incorporating this statistical perspective into fire safety assessments of waste treatment facilities becomes crucial. By considering the potential difficulty and duration of firefighting efforts, appropriate measures can be implemented to enhance fire prevention strategies, improve emergency response plans, and minimize the impact on the facility, the environment, and the surrounding community.

#### 3.4.2. Dispatched Firefighters and Engines

Figure 6 illustrates the correlation between the number of dispatched firefighters and fire engines by different waste treatment facilities and the rescue duration. Notably, "wood-related" and "transport vehicle" fires demonstrate a clear association between the number of responders and the duration of rescue operations. However, for other fire types, the correlation is less pronounced. The analysis revealed that the required number of firefighters and fire engines for each fire type is influenced by factors such as fire location, nearby street conditions, simultaneous fires, and the decisions made by the commanding officer. Therefore, there may be instances where the rescue operation takes a long time despite the relatively lower number of dispatched firefighters and fire engines.

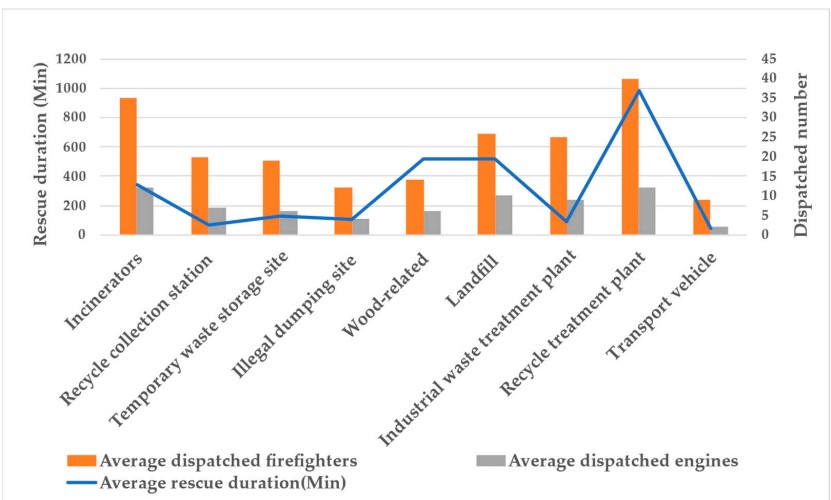

**Figure 6.** Number of dispatched firefighters and engines compared with rescue duration.

### 3.4.3. Affected Area

Figure 7 examines the relationship between rescue duration and the affected area. Surprisingly, the analysis reveals little correlation, which differs from the commonly held notion. While it may be expected that larger affected areas would require longer rescue times, the findings do not align with this expectation. It is important to note that the information regarding the affected area is obtained from the content of rescue reports. During the interviews with firefighters, inquiries were made about how the affected area is determined and whether there is a standardized criterion for assessment. Unfortunately, the response indicated no unified standard; commanders rely on their experience when reporting on and documenting the affected area. Consequently, it becomes difficult to objectively analyze the correlation between the affected area and the scale of the fire based on this information.

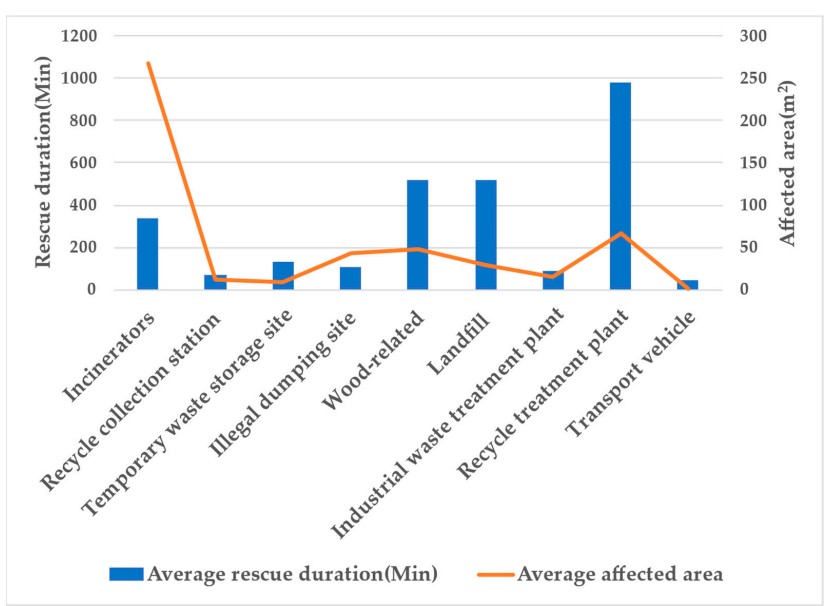

**Figure 7.** Average affected area and relative comparison with rescue duration.

### 3.5. Fire Risk Assessment

Based on the causes, occurrence frequency, fire extinguishing time, and hazard characteristics of fires in different types of waste treatment facilities discussed above, a fire safety assessment table for waste treatment facilities is compiled (see Table 2).

**Table 2.** Fire risk assessment of waste treatment facilities.

| Level of Risk | Type of Facility | Frequency of Fire | Rescue Duration (Min) | Fire Risk | Hazard Characteristics |
|---|---|---|---|---|---|
| 🟥 | Illegal dumping site | Often | 107 | High | Large quantities Pollutants Deep-seated fire |
| 🟥 | Wood-related | Often | 517 | High | Large quantities Deep-seated fire |
| 🟥 | Recycle treatment plant | Regularly | 982 | High | Large quantities Pollutants Damage on equipment |
| 🟧 | Landfill | Very rarely | 522 | Moderate | Large quantities pollutants |
| 🟧 | Incinerators * | Very rarely | 339 | Moderate | Large quantities |
| 🟧 | Recycle collection station | Often | 68 | Moderate | Complex waste types Pollutants |
| 🟨 | Temporary waste storage site | Rarely | 130 | Low | Complex waste types |
| 🟨 | Industrial waste treatment plant | Rarely | 88 | Low | Large quantities Pollutants Damage on equipment |
| 🟪 | Transport vehicle | Very rarely | 43 | Minimal | Moving object |

* Waste collection area.

The evaluation of the frequency of fire occurrence can be categorized as follows:

- Often: This rating indicates that fires frequently occur in specific types of waste treatment facilities. These facilities, including "wood-related" (83 times), "illegal dumping sites" (82 times), and "recycle collection stations" (76 times) have fire incidents beyond or near the third quartile of 79.
- Regularly: This rating signifies that fires in a particular waste treatment facility are ongoing but less frequent than the "often" category. The number of fires exceeds the second quartile by 11 times and remains well below the third quartile of 79. The "recycle treatment plants" fall into this range with 29 recorded fire incidents.
- Rarely: This rating suggests infrequent or sporadic fire incidents in a specific type of waste treatment facility. The number of fires surpasses the first quartile of 4 and is less than or equal to the second quartile of 11. The "industrial waste treatment plant" had 11 fire incidents, while the "temporary waste storage site" had seven incidents within this range.
- Very rarely: This rating indicates uncommon and rarely observed fires in a specific waste treatment facility. The "incinerator" had four fire incidents, while the "transport vehicle" and "landfill" had four and three fire incidents, respectively, falling into this category.

Regarding the degree of fire risk, the comprehensive assessment of fire risk encompasses both the frequency and severity of the fire occurrence. While the frequency aspect has been previously discussed, the severity component poses difficulties in quantification. Therefore, the evaluation relies on factors such as average rescue duration and the combustion characteristics of waste in the facility. The hazard characteristics section specifically focuses on the risk posed to the surrounding environment, society, and residents following a fire incident.

To present the overall assessment, color blocks can be used to indicate the level of risk, ranging from high to minimal risk. The evaluations can be represented as follows:

- High Risk: This rating is depicted in red, indicating a significant level of fire risk. Facilities categorized as high risk may have prolonged rescue durations, highly combustible waste materials, and considerable risks to the surrounding environment, society, and residents in the event of a fire.
- Moderate Risk: This rating is represented by orange, signifying a moderate level of fire risk. Facilities categorized as moderate risk may have moderate rescue durations, com-

bustible waste materials with certain risks, and potential impacts on the surrounding environment, society, and residents in case of a fire.

- Low Risk: This rating is illustrated in yellow, indicating a relatively low level of fire risk. Facilities categorized as low risk may have shorter rescue durations, less combustible waste materials, and minimal risks to the surrounding environment, society, and residents in the event of a fire.
- Minimal Risk: This rating is represented by pink, symbolizing a minimal level of fire risk. Facilities categorized as minimal risk may have swift rescue durations, non-combustible waste materials, and minimal to no risks to the surrounding environment, society, and residents following a fire incident.

Using color blocks and the corresponding evaluations allow for a clear visual representation of the overall fire risk assessment, enabling stakeholders to easily understand the level of risk associated with different waste treatment facilities. It should be noted that the risk assessment is qualitative, and a semi-quantitative approach for analyzing the fire safety of waste facilities will be published in due course.

### 3.6. Fire Safety Measures

Fires in waste treatment facilities have wide-ranging impacts on the environment, society, and economy, and can pose serious health risks to the surrounding residents. As a result, it is imperative to approach fires in waste facilities from a disaster management perspective rather than treating them as general fire incidents [27]. This entails adopting a comprehensive approach encompassing prevention, disaster reduction methods, disaster response, and post-event recovery strategies, incorporating fire management regulatory perspectives [28], insights from frontline firefighters, and new fire technologies [29,30]. Applying the four strategic steps—prevention, disaster reduction, disaster response, and post-event recovery—should extend beyond the daily operations and management of waste treatment facilities. It is crucial to consider these steps in the following fields:

- Design and construction of facilities and factories.
- Management organization and plan.
- Waste storage and treatment.

The measures suggested in Table 3 show a summary of the results of this study.

It is, however, important to note that Table 3 only includes measures for legally registered waste treatment facilities. In addition, a significant percentage of fire incidents occur in illegal dumping sites, which are characterized by remote locations, no buildings, and difficult to monitor, making it challenging to provide specific fire safety recommendations for these sites. In addressing this issue, it becomes crucial to focus on preventive measures through regulatory policies to deter illegal waste dumping. One approach to combating illicit dumping is the utilization of satellite observation technology. This technology enables the regular monitoring of remote areas such as mountains and suburbs to identify locations where illegal dumping occurs. However, it is crucial to recognize that this method primarily focuses on detecting illegal dumping after environmental damage.

Consequently, subsequent waste disposal efforts require significant manpower and material resources to rectify the situation. To effectively address the issue of illegal dumping, it is necessary to implement comprehensive waste management strategies. This involves strict monitoring and tracking of waste flow, ensuring that waste is disposed of through proper channels. In addition, regulatory authorities must remain vigilant to prevent unscrupulous operators from unlawfully dumping waste. By implementing stringent waste management practices and enforcing regulations, it is possible to reduce the occurrence of illegal dumping and its associated fire hazards. This approach requires a collaborative effort between regulatory agencies, waste treatment facilities, and the community to create a culture of responsible waste disposal.

**Table 3.** Recommended safety measures: prevention, disaster reduction, disaster response, and recovery.

| Fields of Action | Strategic Steps | Measures |
|---|---|---|
| Design and construction of facilities and factories | Prevention | • Install adequate waste status monitoring equipment.<br>• Ensure the load margin and protective measures of the power circuit.<br>• Proper ventilation systems. |
| | Disaster reduction | • Ensure space for storage of waste.<br>• Implement fire-resistant materials.<br>• Install proper fire detection and suppression equipment * [29]. |
| | Disaster response | • Proper escape route.<br>• Proper space for emergency actions. |
| | Recovery | • Evaluate the structural properties of buildings and facilities.<br>• Assess the adequacy of fire suppression and monitoring equipment.<br>• Assess the adequacy of space for waste storage, emergency actions, and escape. |
| Management organization and plan | Prevention | • Establish a dedicated management organization and program for fire safety.<br>• Establish SOP for every step of operations.<br>• Regular fire risk assessments. |
| | Disaster reduction | • Develop clear emergency action plans for fire incidents.<br>• Plan for detection of fires, including manual monitoring.<br>• Establish incident reporting and communication systems. |
| | Disaster response | • Provide ongoing training to staff on emergency response and evacuation protocols.<br>• Regular fire drills.<br>• Proper personal protection equipment ** [30]. |
| | Recovery | • Documentation of case history, outcomes, and experience.<br>• Re-assess fire management and response plans. |
| Waste storage and treatment | Prevention | • Adequate storage plan, including how to classify and store different waste types.<br>• Regular waste sampling and monitoring protocols.<br>• Establish complete staff operating regulations and regular education and training. |
| | Disaster reduction | • Limit the size of waste stockpiles and storage time.<br>• Plan fixed storage space for high-risk waste.<br>• Ensure adequate space between waste stockpiles to limit fire spread and extinguish fire. |
| | Disaster response | • Adequate method or equipment of fire extinguishing.<br>• Quick respondents and stop operations.<br>• Ensure the protective equipment.<br>• Evacuation and escape actions when the fire expands. |
| | Recovery | • Proper disposal of contaminated runoff water and fire-damaged waste.<br>• Review and reallocation of the method and space of waste storage and classification. |

* More sensitive fire detection technologies and devices, such as detectors that react at low temperatures or that can transmit wirelessly [29]. ** For example, fire suits that can respond to early fire alarms before they fail under extreme fire conditions [30].

## 4. Conclusions

Based on an analysis of rescue reports, investigative reports, and interviews with firefighters, this study highlights the significant role of "human factors" in waste treatment facility fires. The study also examines the geographical and temporal distribution patterns of these fires. Finally, to enhance risk management and control measures for waste facility fires, the study proposes several recommendations:

- Strengthening Management Practices: Stakeholders should proactively enhance management practices within waste treatment facilities. This includes implementing robust protocols, establishing clear guidelines, and ensuring adequate supervision and oversight. Additionally, stakeholders should consider installing monitoring equipment to detect potential fire hazards and address them promptly.
- Strengthening Supervision: It is crucial to strengthen the enforcement of policies and regulations, promoting compliance among waste facilities. This includes conducting regular inspections, imposing penalties for non-compliance, and providing support and resources for implementing effective fire safety measures.

The continuous increase in waste generation and the rapid development of related industries further heighten fire risks. By combining the insights gained from this study with ongoing research and collaborative efforts, we can strive to improve fire safety at waste treatment facilities and mitigate the environmental, social, and economic impacts of fires. This study serves as a foundation for understanding specific aspects of fires in the waste treatment industry. However, further research is needed to deepen our knowledge and develop comprehensive strategies for effectively addressing this issue.

**Author Contributions:** Conceptualization, methodology, formal analysis, investigation, data curation, and writing—original draft preparation: W.-Y.J.; writing—review and editing: C.-L.W. and F.-W.L.; supervision: C.-L.W., F.-W.L. and W.-S.C. All authors have read and agreed to the published version of the manuscript.

**Funding:** This research received no external funding.

**Institutional Review Board Statement:** Not applicable.

**Informed Consent Statement:** Not applicable.

**Data Availability Statement:** Not applicable.

**Acknowledgments:** Thanks to all the firefighters from Kaohsiung City, especially the fire investigation division, for their detailed reports. Thanks to Wu and Liu for their advice on waste management facilities and paper writing. Special thanks to Wei-Sheng Chen for his guidance and advice. This work was support by the National Science and Technology Council of Taiwan under Grant NSTC 111-2222-E-006-016-MY2.

**Conflicts of Interest:** The authors declare no conflict of interest.

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
