# Peer review of "Fires in Waste Treatment Facilities: Challenges and Solutions from a Fire Investigation Perspective"

_sustainability, doi:10.3390/su15129756_

Round 1

Reviewer 1 Report

This paper discussed the challenges and solutions of fires in waste treatment facilities from a fire investigation perspective, based on the fire accidents in waste treatment facilities in Kaohsiung City within five years.This research topic is very interesting, and some of the views provided by the author have some practical significance. However, the overall innovation of the paper is not strong enough. The following are the specific comments of the review.

1 Line 100-120:The authors list several research suggestions that seem out of place here. The brief description before line 100 does not seem to be able to reach these conclusions, and it is recommended to adjust the place.

2 The regression analysis in Figure 3 is mathematically correct, but what is the science mechanism behind it? Or is it realistic?

3 The form of Figure 4 is not intuitive. It is recommended to change it to a different type of diagram.

4 The conclusion part is a little verbose, and part of the content is not involved in the text analysis of the article.

5 The abstract can also be presented in more condensed and concise language.

6 Please provide more detailed information about the interview approach, such as the number of interviewer and interviewee, the method to select the interviewee and the data process method.

The language is generally good, but some of the statements could be more condensed.

Author Response

We thank you for your tremendous effort spent in the peer-reviewing process. The comments are of great value for us to improve the manuscript. We made revisions to the manuscript as uploaded.

All the changes in the revised manuscript were clearly marked. All modifications are marked in red.

We are looking forwards to receiving positive feedback from your side.

Reviewer 2 Report

This article provides valuable suggestions for strengthening the fire safety of waste treatment facilities and contributes to reducing fire risks. In my view, the manuscript could be accepted for the publication in this esteemed journal after incorporation and revision.

The Suggestions are as follows:

(1) Unfortunately, the author did not provide a complete description of their place of belonging. Since the article aims to explain the issue from the perspective of fire supervision, it is closely related to the characteristics of the region. Therefore, the author should clearly state in which country.

(2) In the introduction, the content of lines 53 to 60 cannot be connected with the previous paragraph and the next paragraph. Logically, it only needs to be said that a fire in a waste treatment facility can cause trouble, so fire safety measures need to be taken as a priority, and there is no need to add a paragraph about "unclear classification of fire in waste treatment facilities". It’s suggested to make modifications to this section.

(3) The logic of the last paragraph of the preface is not clear. Many other researchers' studies on the factors affecting the fire of waste treatment facilities have been cited, but they are not connected with the follow-up "Why should we study the fire of waste treatment facilities in Taiwan, China ". Suggest strengthening the logical connection between paragraphs.

(4) In section 3.5, there is a situation of inconsistent fonts. It is recommended to check the entire text for any issues of the same type.

(5) In section 3.6, it is difficult to provide specific fire safety recommendations for illegal dumping of waste at these locations. The author can refer to the literature “Chemical Engineering Journal 2023, 460: 141661 (https://doi.org/10.1016/j.cej.2023.141661)” and “Chemical Engineering Journal 2022, 431: 134108 (https://doi.org/10.1016/j.cej.2021.134108)” to conduct fire warning and monitoring in these places.

Moderate editing of English language required.

Author Response

We thank you for your tremendous effort spent in the peer-reviewing process. The comments are of great value for us to improve the manuscript. We made revisions to the manuscript as uploaded.

All the changes in the revised manuscript were clearly marked. All modifications are marked in red.

We are looking forwards to receiving positive feedback from your side.
Please see the attachment

Reviewer 3 Report

Many thanks for the opportunity to review this paper on what is a very interesting subject. In the following I give a series of minor editorial comments, and then some deeper comment on suggested additions to make the paper more robust in its analyses.

Editorial comments

·       Add citations about fires described in lines 64-69,

·       Please place citations in the first sentence in which a new source is mention, for example cite Ibrahim et al. at end of the sentence ending in line 73 rather than later in line 79,

·       Please directly reference Figure 1 from a relevant location in the text and be clear in the caption that the map extent is the boundary of Kaohsiung city (if that is the case)

·       Figure 2, axes need labels

·       Figure 4, some of the colours are too similar, please make each ignition source a more obvious colour,

·       Figure 6, please explain the significance of the different blue and orange colour bars,

·       Line 215, please add comma instead of full stop before “as shown in figure 3”

·       Line 252-253 is not correct English, even though I can understand what the authors mean,

·       Line 260 sentence is incomplete, wood fires are what..? I think full stop in line 262 should be a comma instead to make one longer sentence,

·       Line 346, put “see table 2” in brackets, not as a separate sentence

·       Add more lines or shading in table 3 to make it clear which measures fall under which strategic step.

     Analysis comments

I’m not sure there is enough evidence to say rain is more important than temperature, and I am not convinced that the regression in fig 3 is beneficial to the paper. The interaction between temperature, rainfall and seasonal variation in ignition sources due to the various described festival traditions is certainly a complex one, but I recommend you leave this section as qualitative instead of trying to quantify which factor is most important. I recommend removal of Figure 3.

The method states that on-site interviews of fire fighters were used as a cross reference against the fire database. Please indicate in the discussion sections if, when and where information provided by the fire fighters was crucial in completing information that the database lacked – otherwise it is unclear why the interviews were required in the first place.

On fire risk, please be more explicit that there is a difference between risk of fire ignition, and risk of fire severity. The authors comment on fire severity indirectly through Figure 6 and fire duration. However, a great addition to the paper would be to compare fire severity with temperature/rainfall/seasons as the authors already do with ignitions. Are there available data on this beyond fire duration? Maybe fire size? Number of firefighters/fire engines required? Certainly I would expect to see much larger fires on average in hotter/dryer months and this should be easier to quantify than risk of ignition.

Please explain “broken window theory” (line 305) more to the reader as it is unclear what is meant by this. Otherwise I recommend that the reference to it is removed, I don’t think it is necessary to give it a specific name for the reader to understand what the authors are trying to explain here.

Please explain further how unknown causes/cigarettes/other causes have correlation with natural conditions (line 310), e.g. are cigarettes not a human factor?

On fire frequency (starting line 347), since you have actual data of fires and a time span over which they occurred, please provide some numerical context on the different fire frequencies. For example, there were about 80 wood-related fires over the 4 years of data, so by analysis that must be an average of about 20 fires per year which the authors regard as ‘often’. Can you extend this to each category and give an approximate range of number of fires for each one?

Given you state that Table 3 is for legally-registered facilities, I am led to believe this is not information that the authors have themselves produced in this research but is actual legal practice in Taiwan. If so, please cite the source of the information whether that be from legal documents of design codes.

Generally, at no point in the paper am I really convinced of the impacts of fires – there is variously some reference to exposing people to risk and “substantial damage and losses” but some quantitative context would be nice. What about deaths and injuries? Costs of clean-up and business interruption? Financing required to maintain fire service? Any statistics in the introduction would be hugely beneficial to convince the reader of the problem.

Generally fine. Can be understood. One or two erroneous full stops that need replaced with commas to make complete sentences.

Author Response

(The authors gave the same response as above.)

Reviewer 4 Report

General: The author employed three approaches to gain a comprehensive understanding of waste facility fires in Kaohsiung City, contributing to the development of effective fire prevention, response, and mitigation measures. To accept, the following secondary issues need to be addressed. In addition, additional proofreading for grammar and spelling is required.

1.       It is recommended to refrain from using Excel for drawing graphs and instead utilize professional graphing software for graphic illustrations.

2.       It is recommended to zoom in on the densely populated area in Figure 1.

3.       The inconsistency in the font of "Figure 5 The statistic of fire cause" has been noted. It will be formatted to match the style of other figures for consistency.

4.       The inconsistency in formatting between figure titles and table titles has been noted, with some titles being bolded and others not. Please modify.

5.       The inconsistency in font styles between the images and tables should be addressed and rectified accordingly.

6.       It is recommended to reduce the number of words in the abstract and summarize the main points of the article in more concise language.

7.       It is suggested to improve the match rate of references and provide explanations in the article for similar references.

8.       The title of the article is not logical enough. It is suggested to revise the title to enhance the logic and make the content of the article more reasonable and reliable.

Minor editing of English language required

Author Response

(The authors gave the same response as above.)

Round 2

Reviewer 2 Report

The author has made revisions based on the comments of the reviewers, and I believe that this manuscript can be published in this respected journal.